nanotechnology/biochemistry/biophysics

nanocomposites, magnetic cores, silver nanoparticles, cytotoxicity, antibacterial activity

**Author for correspondence:**
Muhammad Akram Raza
e-mail: akramraza.cssp@pu.edu.pk

This article has been edited by the Royal Society of Chemistry, including the commissioning, peer review process and editorial aspects up to the point of acceptance.

# Synthesis and characterization of silver nanoparticle-decorated cobalt nanocomposites (Co@AgNPs) and their density-dependent antibacterial activity

Zakia Kanwal[1], Muhammad Akram Raza[2], Saira Riaz[2], Saher Manzoor[2], Asima Tayyeb[3], Imran Sajid[4] and Shahzad Naseem[2]

[1]Department of Zoology, Lahore College for Women University, Jail Road, Lahore 54000, Pakistan
[2]Centre of Excellence in Solid State Physics, [3]School of Biological Sciences, and [4]Department of Microbiology and Molecular Genetics, University of the Punjab, Quid-e-Azam Campus, Lahore 54590, Pakistan

ZK, 0000-0002-0539-3056; MAR, 0000-0002-7895-5013

Magnetic cores loaded with metallic nanoparticles can be promising nano-carriers for successful drug delivery at infectious sites. We report fabrication, characteristic analysis and *in vitro* antibacterial performance of nanocomposites comprising cobalt cores (Co-cores) functionalized with a varied concentration of silver nanoparticles (AgNPs). A two-step polyol process synchronized with the transmetalation reduction method was used. Co-cores were synthesized with cobalt acetate, and decoration of AgNPs was carried out with silver acetate. The density of AgNPs was varied by changing the amount of silver content as 0.01, 0.1 and 0.2 g in the synthesis solution. Both AgNPs and Co-cores were spherical having a size range of 30–80 nm and 200 nm to more than 1 μm, respectively, as determined by scanning electron microscopy. The metallic nature and face-centred cubic crystalline phase of prepared nanocomposites were confirmed by X-ray diffraction. Biocompatibility analysis confirmed high cell viability of MCF7 at low concentrations of tested particles. The antibacterial performance of

nanocomposites (Co@AgNPs) against *Escherichia coli* and *Bacillus subtilis* was found to be AgNPs density-dependent, and nanocomposites with the highest AgNPs density exhibited the maximum bactericidal efficacy. We therefore propose that Co@AgNPs as effective drug containers for various biomedical applications.

# 1. Introduction

Nanotechnology has revolutionized almost every field of human life owing to the unique and astonishing physiochemical, electrical and mechanical properties of nano-sized materials. The quantum confinement effects and large available active surface areas are believed to be the key factors to the enhanced functionality of nanostructures. These properties make them suitable for various biomedical applications, e.g. targeted drug delivery to improve therapeutic efficacy, cellular repairing, gene therapy, diagnosis at single cell and molecular level integrated therapeutics and progress in diagnostic tests and medical devices [1–3].

Perhaps antibiotics seem the only option against the life-threatening pathogenic bacterial infections and since the invention of penicillin, antibiotics have saved countless lives [4]. However, many microorganisms have developed resistance against current drugs to become 'superbugs'. There are numerous reasons for this resistance such as extensive use of these antibiotics, wrong prescriptions, low quality, lack of discovery of new drugs, poor hygienic conditions and use of antibiotics in animals and crops [5]. A high percentage (more than 60%) of nosocomial infections are caused by these 'superbugs' and use of more antibiotics is producing adverse side effects making this challenge more serious for global healthcare [5–8]. This scenario demands novel strategies to fight against resistive pathogens and the nanoparticles can be a promising approach [4,6–9].

Silver, owing to its bactericidal nature, has been used for medical purposes since ancient times and highly reactive surfaces of silver nanoparticles (AgNPs) suggest their potential role in antimicrobial applications [10,11]. Different possible antimicrobial mechanisms could be involved in the microbial killing action of AgNPs such as $Ag^+$ ion release, DNA damage, cell membrane disruption and electron transport. By virtue of broad-spectrum killing, low toxicity to the environment and better oligodynamic performance, AgNPs are highly rated antimicrobial agents used in commercialized consumer products. The list of AgNP-based products includes portable clean water filters, diabetic foot and wound dressings, antimicrobial coatings on surgical instruments and devices, antibacterial soaps, skin lotions and creams [12–16].

Although AgNPs are almost non-toxic at low concentration, the accumulation of AgNPs in mammalian cells may cause side effects and infections such as argyrosis and argyria by interaction with different organelles and subcellular components of the body [11,17,18]. Therefore, the removal of Ag nanoparticles from the body is very essential and a serious challenge. Furthermore, the use of smaller NPs which are considered more effective bactericidal agents can enhance the tendency of potential cytotoxic effects in a human cell by penetration [8]. In order to cope with this problematic scenario, different new strategies are required and loading of AgNPs on magnetic cores to synthesize nanocomposites could be one of them. The magnetic core-based nanocomposites can support to realize the effective recovery of the residual particles from the medium after successful application. Moreover, the decoration of AgNPs on the magnetic particles can also provide stability to their magnetic dispersion [7,19,20]. Magnetic materials, by virtue of their biocompatibility, low toxicity and capability to transform electromagnetic energy to heat, can be employed in different clinical procedures, e.g. magnetic resonance imaging (MRI), hyperthermia, biomagnetic separation, drug delivery and cancer therapy [21–24].

Among the magnetic materials, Iron (Fe), Cobalt (Co) and Nickel (Ni) are considered the most prominent and attractive ferromagnetic elements which belong to the 3d-block. Owing to their excellent magnetic characteristics in the elemental forms, they are widely used as magnetic cores for the fabrication of nanocomposite structures. For Fe, Co and Ni, the saturation magnetization (Ms) values are 220, 170 and 55 emu $g^{-1}$, respectively, at room temperature while Curie point (TC) values are 770°C, 1131°C and 358°C, respectively. Although the higher Ms value of Fe makes it superior in magnetization in comparison to Co and Ni, the larger value of negative reduction potential (E0) proves it unstable due to fast oxidation in air. This oxidation process at nanoscale further enhances due to larger surface to volume ratio making Fe nanoparticles highly prone to oxidation as compared to the other two elements. In the comparison of Co and Ni, somewhat higher Curie point, larger

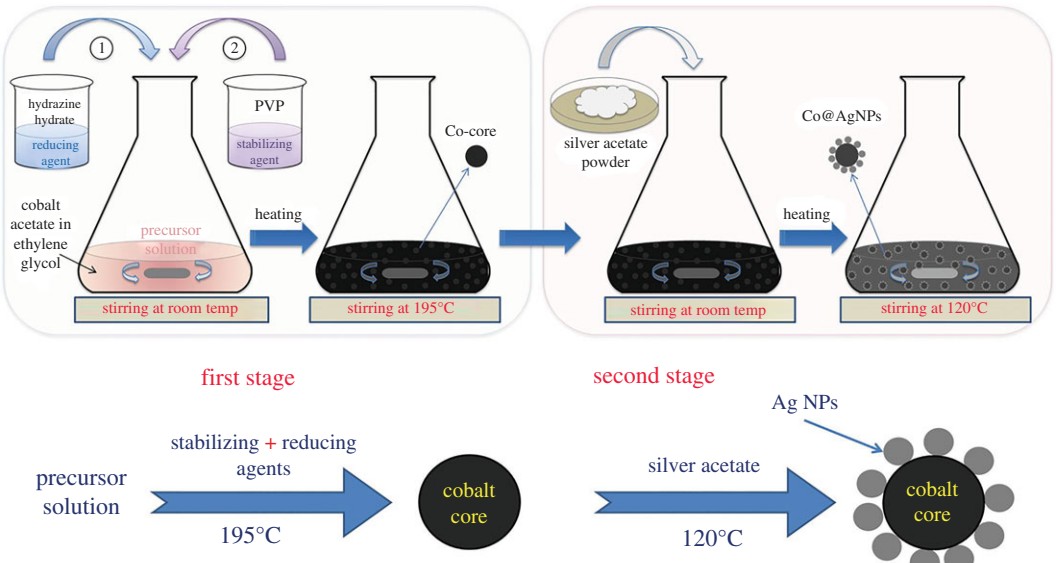

**Figure 1.** Schematics of the nanocomposite preparation routes. First, synthesis of Co-cores was carried out, followed by the fabrication of AgNPs on them.

magnetization and greater magnetocrystallinity make Co an ideal core material for hybrid nanostructures. Various nanocomposite materials including core−shell structures have been reported using Co as the core material, for example, cobalt−gold (Co@Au), cobalt−copper (Co@Cu), cobalt−platinum (Co@Pt) and cobalt−silver (Co@Ag) nanoparticles [25−27]. The prominent advantages of using Co as a core material with other non-ferromagnetic materials such as Ag, Au and Cu include giant magnetoresistance (GMR) and high stability even at higher temperatures, which are desirable properties in multifunctional sensing, media recording and wideband photovoltaic solar cell applications [28−31]. Numerous chemical, physical and biological approaches are used to synthesize bimetallic hetero-nanostructures; however, solution phase chemical reduction approach can be advantageous due to its simple handling and cost-effectiveness. Besides, it does not require any sophisticated equipment and provides high purity index, thermal stability and control over tailoring the properties of prepared nanostructures by tuning particle size [25,32,33].

In this work, we fabricated AgNPs on magnetic Co-cores to manufacture bimetallic nanocomposites (Co@AgNPs). The variation in the density of deposited AgNPs was accomplished by changing the amount of silver precursor in the synthesis solution. Controllable active surface areas, tuneable antibacterial properties and magnetic nature of manufactured nanocomposites indicate their potential roles for the biomedical application.

# 2. Material and methods

## 2.1. Material and chemicals

All the chemicals including cobalt acetate tetrahydrate (Co(CH₃COO)₂·4H₂O), silver acetate (CH₃COOAg), ethylene glycol (C₂H₆O₂), polyvinylpyrrolidone (PVP) and ethanol were of research grade and purchased from Merck (Germany). Hydrazinium hydroxide 80% PA (N₂H₄·H₂O) was from Panreac, Barcelona, Spain. The deionized (DI) water was used for synthesis, solution making and other purposes throughout the experiment. Both bacterial strains (*Escherichia coli* and *Bacillus subtilis*) were provided by the Microbiological Department of our university.

## 2.2. Preparation of silver nanoparticle-decorated cobalt nanocomposites

The preparation of bimetallic composite nanostructures was carried out in two steps as shown in figure 1, by employing the polyol process with transmetalation reduction mentioned in detail previously [25] with some modifications as described below briefly. A quantitative summary of chemicals used and the experimental conditions at each step is described in table 1.

**Table 1.** Details of chemicals and experimental conditions for the preparation of nanocomposite structures (Co@AgNPs).

| sample | precursor | reducing agent | stabilizing agent | reaction conditions |
|---|---|---|---|---|
| S0 | cobalt acetate (48 mM, 25 ml) | hydrazine hydrate (0.4 ml), ethylene glycol (25 ml), added dropwise | PVP (0.25 mM, 50 ml) | heated up to 195°C under refluxing conditions with constant stirring (100 r.p.m.) |
| S1 | silver acetate (0.01 g) added to S0 | — | — | mixed by stirring (100 r.p.m.) at room temperature |
| S2 | silver acetate (0.1 g) added to S0 | — | — | mixed by stirring (100 r.p.m.) at room temperature |
| S3 | silver acetate (0.2 g) added to S0 | — | — | mixed by stirring (100 r.p.m.) at room temperature |

### 2.2.1. Synthesis of Co-cores

In the first step, Co-cores were synthesized by using the precursor solution (48 mM) of cobalt acetate dissolved in ethylene glycol (EG). Hydrazine hydrate served as the reducing agent and PVP served as the stabilizing agent. The overall reaction was carried out under refluxing conditions with constant stirring (100 r.p.m.). To 25 ml of precursor solution, a mixture solution of 0.4 ml of hydrazine hydrate and 25 ml of ethylene glycol was added dropwise under constant stirring at room temperature. Then 50 ml of a polymer solution (0.25 mM), prepared independently by dissolving PVP into EG was added and stirred for 15 min. The solution was heated up to 195°C and the temperature was maintained for half an hour. The solution turned to black confirming the completion of the reaction. The Co-cores (sample S0) were collected magnetically for further characterization.

### 2.2.2. Fabrication of AgNPs on Co-cores

In the next step, decoration of AgNPs on these pre-fabricated Co-cores was achieved by using three amounts of silver acetate powder: 0.01, 0.1 and 0.2 g, and obtained samples were labelled as S1, S2 and S3, respectively. Typically, a predetermined amount of silver acetate was mixed mechanically (stirring at 100 r.p.m.) for 15 min at room temperature into a colloidal solution of Co-cores. The solution was heated up to 120°C and kept for 15 min under continuous stirring. Afterwards, prepared nanocomposites were obtained using permanent magnets and washed several times with ethanol before further applications.

## 2.3. Characterization of prepared nanocomposites

Different characterization techniques were used for morphological, optical, magnetic and structural analyses of the prepared samples.

For structural investigations, an X-ray diffractometer (D-maxIIA, Rigaku, Japan) operated at 40 kV and 25 mA using Cu K$\alpha$ line radiation ($\lambda = 1.5406$ Å) was used. A thick film of each sample was achieved on a clean glass substrate by drying the colloidal sample for XRD measurements. Scanning electron microscopy (Nova NanoSEM 450, USA) was carried out at different magnifications for morphological investigations. Again drop casting approach was employed to achieve enough material of each sample on glass substrates for SEM analysis. A thin gold layer (10 nm) was also deposited, if needed, to avoid any possible charging on samples. Optical response of all colloidal samples was determined by ultraviolet–visible spectroscopy (Nicolet, Evolution 300, USA) and absorbance spectra

were measured in the wavelength range from 300 to 700 nm. To determine the magnetic properties, hysteresis curves for each sample were obtained by using the vibrating sample magnetometer (VSM: 7407, Lakeshore, USA)

## 2.4. Cell cytotoxicity assay

In order to determine the toxic effects of the prepared samples, cell cytotoxicity assay was conducted using the human breast cancer cell line, MCF7. A Dulbecco's modified Eagle medium was prepared by using 10% fetal bovine serum, 100 units ml$^{-1}$ penicillin and 100 µg ml$^{-1}$ streptomycin in which cells were cultured. Incubation was done in a humidified controlled atmosphere at 37°C under 5% $CO_2$. Media were routinely changed every third day.

MTT (3-(4, 5-dimethyl-2-thiazolyl)-2,5-diphenyl-2-H-tetrazolium bromide) assay was carried using a standard protocol. Briefly, cells were cultured in a 96-well plate at a density of $10 \times 10^3$ cells ml$^{-1}$. After 24 h, prepared NPs in varying concentrations from 0 to 20 µg ml$^{-1}$ were prepared in culture media and each concentration was subjected to cells in triplicate wells for 72 h. After 72 h, plates were scanned for the background absorbance of NPs for each well; 10 µl of MTT solution per well was added and allowed to crystallize at 37°C for 4 h; 100 µl of acidified isopropanol (as solubilization solution) was then added in each well. Solution was thoroughly mixed to properly dissolve formazan crystals. Absorbance was measured at 570 nm with ELIZA reader.

## 2.5. Antibacterial activity tests

The antibacterial properties of Co-cores (S0), as well as all AgNP-decorated composite structures (S1, S2 and S3) were tested for *E. coli* and *B. subtilis* by Kirby Bauer disc diffusion method [34]. All experiments were performed in triplicate.

The protocol followed for bactericidal activity tests is briefly described: *E. coli* and *B. subtilis* were freshly grown overnight in liquid broth. The concentration of *E. coli* and *B. subtilis* was $5.5 \times 10^8$ and $8.3 \times 10^8$ colony forming units (CFU) ml$^{-1}$, respectively, measured at 600 nm by a spectrophotometer (IRMECO model 2020). A 100 µl sample of each bacterial suspension cultured in LB (Luria-Bertani) broth was spread and dried on the nutrient agar plates. Aqueous solutions with a constant concentration of 10 µg ml$^{-1}$ for all samples (S0, S1, S2 and S3) were prepared. Sterile paper discs (5 mm) were soaked in colloidal solutions as well as in pure water (negative control) and were kept on the agar plates. After 24 h of incubation at 37°C, zone of inhibition (ZOI), defined as the distance from the edge of the disc to the bacterial lawn, for each sample was measured.

## 2.6. Statistical method

Data are presented as mean value and the standard error of mean (mean ± s.e.); graphs were made in Microsoft Excel program. Significant differences between 0 µg ml$^{-1}$ and other concentrations (5–20 µg ml$^{-1}$) were calculated using GraphPad Prism (v. 7.03, San Diego, CA, USA).

# 3. Results and discussion

## 3.1. Synthesis and characterization of bimetallic nanocomposites (Co@AgNPs)

In figure 2*a*–*e*, different colours of the solution are manifested that appeared during the synthesis of core particles. Occurrence of different colours indicated different phases of the reactions. On dissolving the cobalt acetate in EG, the solution acquired a pink colour (figure 2*a*). Upon addition of reducing and stabilizing agents, the colour of the solution turned from pink to milky pink which then turned to purple/dark purple and then finally black with the heat treatment (figure 2*b*–*e*). This indicated the completion of the reaction. Mixing of silver acetate powder into colloidal cores did not produce any major change in colour; it only turned the blackish colloidal solution into greyish silver colour.

During the synthesis of Co-cores, metallic precursor (cobalt acetate) provided the $Co^{2+}$ ions, making the solution pink. The introduction of the reducing agent (hydrazine) reduced these $Co^{2+}$ ions into free cobalt atoms ($Co^0$) by gaining the required electrons and changing the colour of the solution to milky pink. Upon adding the stabilizing agent (PVP) and providing enough heating, nucleation and growth

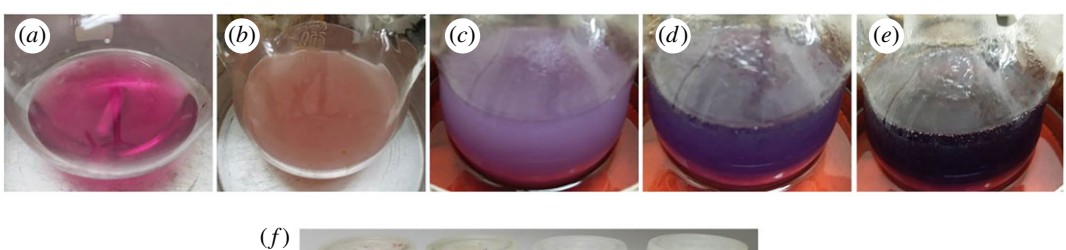

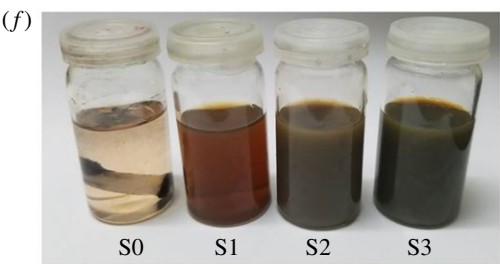

**Figure 2.** (a−e) Manifestation of colour changes that occurred during the synthesis of Co-cores indicating different stages of the reaction; (a) precursor solution, (b) after the addition of reducing and stabilizing agents, (c−e) upon heating the solution up to 195°C. (f) Demonstration of different colours of final colloidal samples; (S0) no silver content, (S1) 0.01 g silver content, (S2) 0.1 g Silver content and (S3) 0.2 g silver content.

processes occurred and free Co-atoms accumulated into Co-particles. Colour changing from milky pink to purple to dark purple and finally black was a manifestation of the various stages of nucleation and growth processes [25]. The addition of silver acetate produced $Ag^+$ ions which were converted into Ag-atoms ($Ag^0$) in the presence of Co-cores. Cobalt particles acted as an electrode during this reduction process and finally converted into $Co^{2+}$ ions again due to the oxidization of Co-atoms of cores. These re-generated $Co^{2+}$ ions may lower the Co-content in the solution causing a reduction in the size of Co-cores. Because of the availability of promising sites on the Co-particles, a heterogeneous nucleation mechanism might occur making a way to the formation of AgNPs on the Co-cores [25,26,35]. Reduction of $Ag^+$ ions into metallic AgNPs on magnetic cores by virtue of anchoring sites was also reported by other researchers [7,20]. The final colours of all prepared colloidal samples are shown in figure 2f.

## 3.2. Structural analysis

The structural properties of all samples were studied by XRD and the results are shown in figure 3. Indexed XRD pattern of Co-cores (S0) without any silver content is represented by a black line in figure 3. The bigger peak showed up at $2\theta = 44.6°$ and one small peak appeared at $2\theta = 75.9°$. These two peaks can be attributed to the (111) and (220) reflection planes, respectively. This spectrum indicated that synthesized Co-cores were of pure metallic and face-centred cubic (FCC) crystalline nature. This is in accordance with the Joint Committee on Powder Diffraction Standards (JCPDS) file no. 15-0806 [36,37].

The XRD pattern of sample S1 is shown by the green line in figure 3. As previously noted, the bigger peak of cobalt appeared at $2\theta = 44.6°$. No peak for silver was noted. The absence of any evident silver peak might be due to the lower concentration of silver content (AgNPs) deposited on the Co-cores [38]. However, a decrease in intensity of the bigger peak (111) at $2\theta = 44.6°$ might be due to the presence of AgNPs on the Co-cores. In figure 3, the indexed diffraction spectra of sample S2 and S3 are represented by blue and red lines, respectively. In both cases, four major peaks were found at $2\theta$ values of $38.4°$, $44.6°$, $64.8°$ and $77.8°$. These characteristic peaks could be attributed to reflection planes (111), (200), (220) and (311) of FCC structure of pure metallic silver according to JCPDS file no. 04-0783 [2,34]. However, the characteristic peak of cobalt (at $2\theta = 44.6°$) was absent. This could be due to the fact that exactly at the same $2\theta$ value (44.6°), a strong silver peak (200) appeared in both cases. Perhaps due to overlapping of both peaks, the cobalt peak could not be distinguished. The increased intensity ($I_{S3} > I_{S2} > I_{S1}$) of the major silver peak (111) reflected the improvement in crystallinity of the deposited material which might be due to the enhanced density of decorated crystalline AgNPs on the Co-cores [39,40]. Thus, XRD results confirmed the bimetallic (cobalt and silver) crystalline nature of our fabricated hetero-structured nanocomposites.

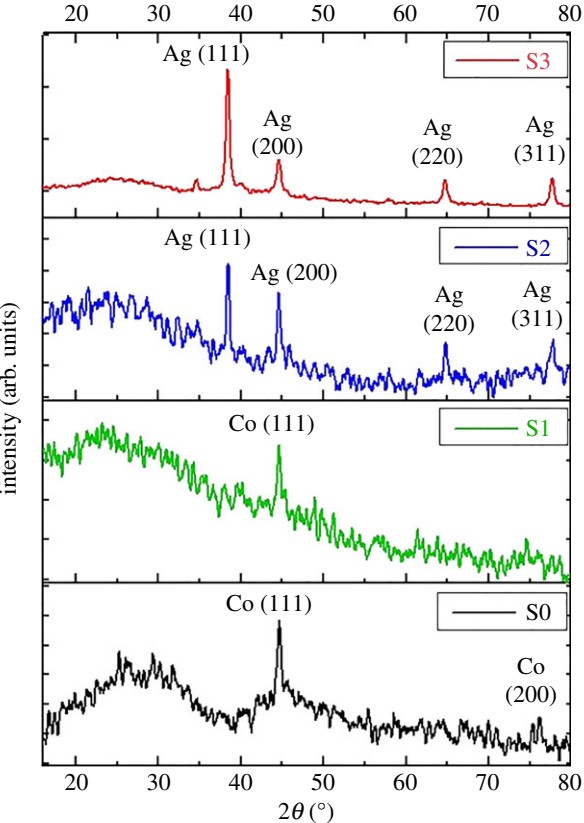

**Figure 3.** XRD spectra of samples S0, S1, S2 and S3 indicating FCC crystalline nature of Co-cores as well as AgNPs. The effect of increasing AgNPs density is also evident.

**Table 2.** Calculated average crystallite size of all samples.

| sample | peak position ($2\theta$) | diffraction plane (hkl) | FWHM (rad) | crystallite size ($D$) (nm) |
|--------|---------------------------|--------------------------|------------|------------------------------|
| S0 | 44.6° | (111) | 0.012 | 11 $\pm$ 1 |
| S1 | 44.6° | (111) | 0.011 | 12 $\pm$ 1 |
| S2 | 38.4° | (111) | 0.0069 | 21 $\pm$ 1 |
| S3 | 38.4° | (111) | 0.0076 | 20 $\pm$ 1 |

The increase in major diffraction peak (111) was used to calculate the crystallite size of each sample with the Debye–Scherer formula [41],

$$D = \frac{k\,\lambda}{B_{\mathrm{hkl}}\cos\theta},$$ (3.1)

where $\lambda$ is the wavelength of X-ray used (1.54 Å), $k$ is the shape factor (0.9), $\beta_{\mathrm{hkl}}$ (taken in radians) is full width half maximum (FWHM) of diffraction peak and $\theta$ is the diffraction angle at that intensity (table 2). A slight change in the crystallite size of samples S0 and S1, as narrated in table 2, shows a negligible reduction in the size of Co-cores when a small amount of silver was deposited. The difference in measured crystallite size (in the case of samples S2 and S3) was not significant. This indicates that a higher amount of silver (0.1–0.2 g) was employed to enhance its density on Co-cores instead of enlarging the size of the particles.

## 3.3. Morphological analysis

The morphological study was performed by SEM and obtained micrographs are presented in figure 4. In figure 4a,b, SEM images of Co-cores are shown at different magnifications. It can be seen that core particles were of spherical shapes with size variation from 200 nm to more than 1 μm. The big size variation could be due to agglomeration (due to the magnetic nature of cobalt) of individual cobalt particles or subunits

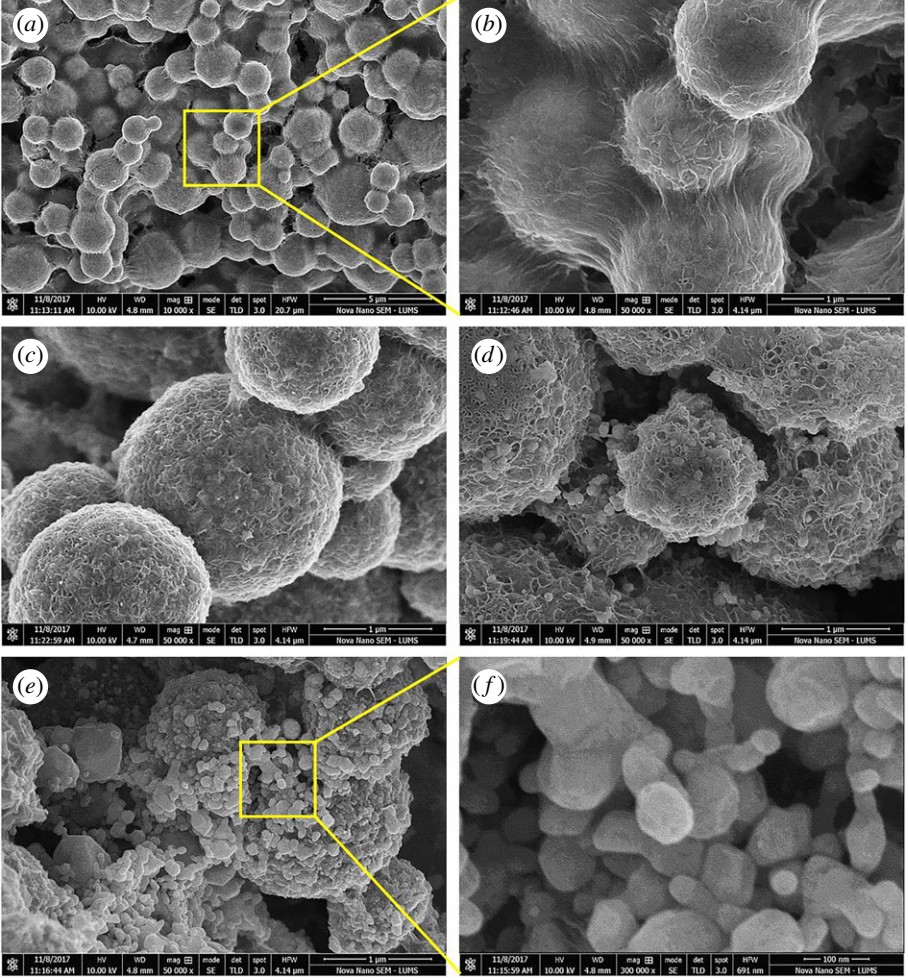

**Figure 4.** SEM micrographs showing the spherical morphologies and size range of cobalt cores sample S0 (*a,b*) and AgNPs in samples of S1 (*c*), S2 (*d*) and S3 (*e,f*).

into larger particles. As magnetic NPs had high surface energy and in order to minimize their energy, they could agglomerate to form larger secondary particles under the effect of magnetization.

The degree of agglomeration depends upon various reaction parameters including the amount of reducing agent. Fabrication of such larger cobalt particles and agglomerated structure using hydrazine as a reducing agent at different temperature was also reported in other studies [35,42,43]. Furthermore, agglomeration of Co-particles could also occur during the deposition and drying process for SEM sample preparation. A close observation (figure 4*b*) revealed the bridge formation between the cobalt particles which could be the result of gold coatings on the sample.

The morphological results for the sample S1 are displayed in figure 4*c*. This SEM image suggests that silver content was deposited on the Co-cores in the form of shells as nano-sized structures (30–50 nm). By increasing the amount of silver to 0.1 g (sample S2), these tiny structures appeared as proper spherical AgNPs having diameters from 30 to 80 nm on Co-cores and their concentration also appeared to increase as is evident in figure 4*d*. In the case of sample S3 (figure 4*e,f*), it can be observed that an increased amount of silver raised the density of AgNPs on Co-particles. A variation in the morphology of silver NPs can also be noted in the magnified view (figure 4*f*). Thus, SEM micrographs clearly showed the decoration of AgNPs on Co-cores and their enhanced concentration with increasing the amount of silver precursor in the solution.

## 3.4. Optical characteristics analysis

The optical responses of all prepared colloidal samples were determined by UV–vis spectroscopy (figure 5). The absorption spectrum of Co-cores sample, shown by the black line, appeared almost

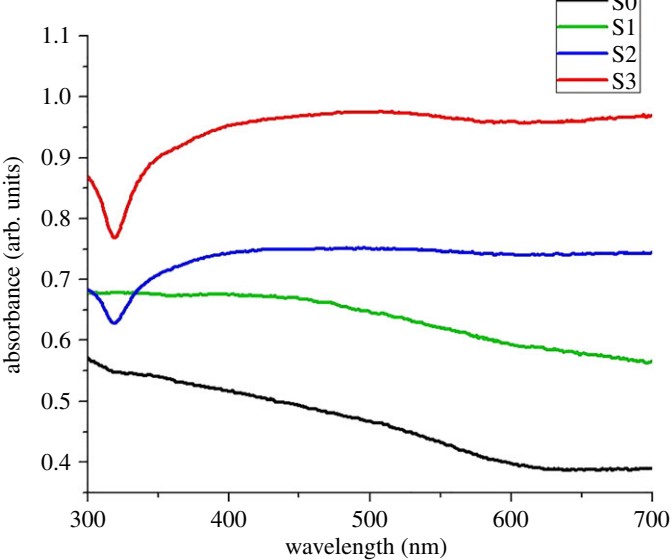

**Figure 5.** UV–vis absorption spectra of all prepared colloidal samples differentiating the samples with a varied concentration of AgNPs.

featureless. The absence of any significant peak at around 350 nm confirmed the pure metallic nature of cobalt particles without any oxides. The monotonic increase in the absorbance along the decreasing wavelength side especially after 600 nm is the characteristic of pure cobalt according to already reported literature [26,44]. The absorption band of sample S1, shown by the green line in figure 5, also revealed a similar response, as in the case of S0, except for a broad bump between 400 and 500 nm possibly due to the presence of a lower amount of deposited silver content. However, the absorbance bands of sample S2 and S3 (denoted by blue and red lines, respectively) in figure 5 exhibited a different trend to sample S0. The broad bands starting from 330 nm and centred at 430 nm can be attributed to the presence of AgNPs on the Co-cores. This band broadening may be ascribed to the size distribution of both core and decorating NPs, synergetic phase retardation effects and electron scattering at the sharing interfaces [31,45]. Similar broad absorbance bands without any sharp shoulder were reported by Lee & Koo for AgNPs decorated on $Fe_3O_4$ cores [46] while Fageria et al. [47] also observed the band widening and reduction in the sharpness of the absorbance peaks for ZnO/Ag and ZnO/Au nanoparticles. In the case of NP-decorated core configurations, the lack of uniformity in deposited material on the cores unlike the core–shell structures may also contribute to the band broadening [48] and thus causing the reduction of peak sharpness. The occurrence of these broad bands without sharp silver peaks may indicate that all AgNPs were deposited on the Co-cores leaving no independent AgNPs in the solution, as reported by Fageria et al. [47] in the case of ZnO/Ag and ZnO/Au nanoparticles. Thus, UV–vis absorption results also confirmed the presence of AgNPs with different concentrations on the Co-cores.

## 3.5. Analysis of magnetic properties

The magnetic behaviour of all four samples was studied by VSM. Magnetizations (M) versus magnetic field (H) results (M–H hysteresis loops) are exhibited in figure 6. Table 3 shows the measured magnetic parameter such as saturation magnetization (Ms), retentively (Mr) and coercivity (Hc). In figure 6, nice S-shaped M–H curves indicate the ferromagnetic behaviour of all tested samples (S0, S1, S2 and S3).

The maximum magnetization (Ms) value for prepared Co-cores was 119.26 emu $g^{-1}$, lower than the bulk cobalt 168 emu $g^{-1}$. However, Hc (98.3 Oe) is much higher than that of bulk (10 Oe) [43,49]. Two main types of effects are considered responsible for such differences in magnetic properties of nanoparticles as compared to their bulk materials [50,51]: (i) size effects due to the differences in single-domain or multi-domain structures along electrons' quantum confinement; (ii) surface effects caused by different defects such as lattices disorder, atomic vacancies, dangling bonds and variations in the atomic coordination that result in surface-spin disorder. A comparison of the hysteresis loops for samples S1, S2 and S3 reveals that Ms values decreased regularly from 110.96 to 90.5 to

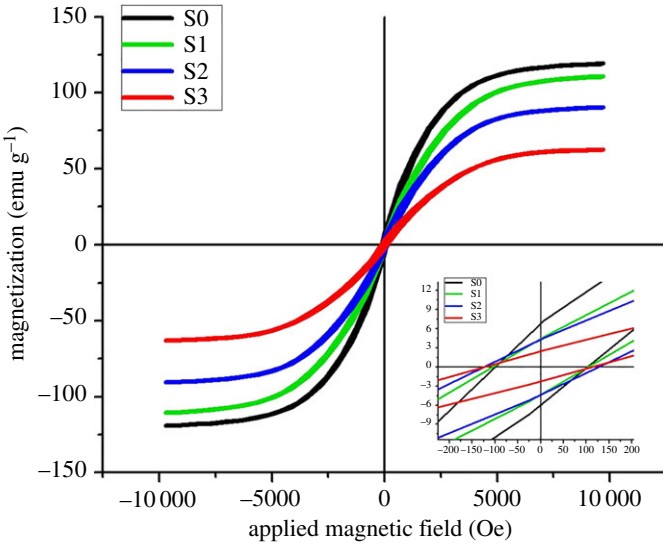

**Figure 6.** VSM magnetic hysteresis curves showing the magnetic properties of Co-cores before and after AgNP deposition.

**Table 3.** Magnetic properties measured from VSM spectra for each sample.

| sample | saturation magnetization (Ms, emg g$^{-1}$) | retentively (Mr, emg g$^{-1}$) | coercivity (Hc, Oe) |
|---|---|---|---|
| S0 | 119.26 | 6.81 | 98.3 |
| S1 | 110.96 | 4.48 | 104.0 |
| S2 | 90.50 | 4.25 | 122.2 |
| S3 | 62.58 | 2.44 | 125.7 |

62.58 emu g$^{-1}$ and Hc increased from 106.2 to 122.2 to 125.7 Oe as the Ag-content was increased from 0.01 to 0.1 to 0.2 g.

This decrease in Ms and increase in Hc of Co-cores by the decoration of the AgNPs can be explained by considering the non-magnetic nature of AgNPs that could develop a strong pinning barrier. The increased amount of Ag-content further enhanced the concentration of AgNPs on Co-cores, thus intensifying the pinning effect [25,31]. Similar variations in magnetic properties (decrease in Ms and increase in Hc) of magnetic cores by capping or decorating with various non-magnetic materials have also been reported in other studies [25,49,52–54].

It is clear from figure 6 that, although magnetic behaviour of Co-cores is compromised by enhancing the Ag-content, nanocomposite structures still retained magnetic properties and can be used for different potential applications.

## 3.6. Cell cytotoxicity analysis

The *in vitro* biocompatibility of the nanoparticles at varying concentration was verified by the MTT assay. The obtained results are demonstrated in figure 7. It can be seen that all the samples were non-toxic up to 10 µg ml$^{-1}$ with cell viability values of more than 70% (74.45 ± 2.3%, 90.59 ± 1.0%, 81.83 ± 2.5%, 75.06 ± 1.4% for S0, S1, S2 and S3, respectively, figure 7). However, at higher concentrations (up to 20 µg ml$^{-1}$), samples were moderately toxic (figure 7). A similar trend of increasing cytotoxicity of CoNPs against human cell line U937 by increasing particles concentration was reported by Nyga *et al.* [55]. They observed cell viability of about 75% at CoNP concentration of 5 µg ml$^{-1}$ which reduced with enhancing the NPs concentrations. Feng *et al.* [56] also noted a concentration-dependent biocompatibility of CoNPs against HIF-1$\alpha$ (+/+) and HIF-1$\alpha$ (−/−) cells. Perhaps at lower concentrations, the amount of released Co ions was not high enough to significantly affect the cell viability.

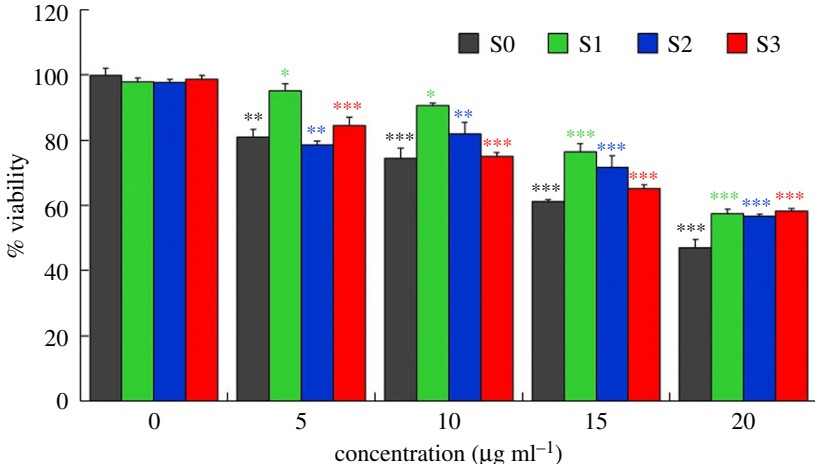

**Figure 7.** Cytotoxicity responses of prepared particles against MCF7 cell line showing concentration-dependent cell viability. Asterisks indicate significant differences between 0 $\mu g\ ml^{-1}$ and higher concentrations (5–20 $\mu g\ ml^{-1}$) of each group (S0, S1, S2 and S3; *$p < 0.05$; **$p < 0.01$; ***$p < 0.001$).

It can also be seen from figure 7 that cell viability improved after the deposition of AgNPs on the Co-cores revealing that nanocomposite particles were even less toxic than Co-core. This reduced toxicity of Co-cores by AgNPs could be explained by considering the lowered release of Co ions by the presence of AgNPs. Nevertheless, a change in cell viability by enhancing the density of AgNPs indicates the variation in toxic effects of nanocomposite particles. The dose and concentration-dependent cytotoxicity of AgNPs against many cell lines have also been described in other studies showing that cell viability decreases by increasing the AgNPs concentrations [57–59].

## 3.7. Antibacterial activity of Co@AgNPs nanocomposites

### 3.7.1. Evaluation of antibacterial efficiency

Bactericidal effects of all synthesized particles against tested microorganisms (*E. coli* and *B. subtilis*) were determined by the aforementioned disc diffusion method. Sterile paper discs soaked in all prepared colloidal samples were positioned on the *E. coli* and *B. subtilis* coated nutrient agar plates. The resulting growth of each bacterium was examined after the incubation period of 24 h at 37°C. Figure 8*a*,*b* shows the representative images of agar plates showing the growth of both bacterial strains, *E. coli* (figure 8*a*) and *B. subtilis* (figure 8*b*) in the presence of tested samples labelled as S0, S1, S2, S3 and pure water (negative control) as Con. The ZOI around each disc was measured in both cases and is listed in table 4. A graphical demonstration of the measured results is shown in figure 8*c*,*d*.

The antibacterial results indicate that each sample exhibited biocidal performance against both strains. The sample S0 (Co-particles with no silver content) showed minimum antibacterial activity in both cases and the measured ZOIs were 1.12 and 0.76 mm, respectively. The ZOI of samples S1, S2 and S3 against *E. coli* were measured as 1.37, 1.86 and 8.51 mm, respectively, while against *B. subtilis*, these values were 0.89, 1.28 and 4.41 mm. It can be observed that the bactericidal performance of synthesized samples against both bacterial strains increased from minimum to maximum as the silver content was enhanced from no silver (S0) to the maximum (S3). However, all samples appeared less effective against *B. subtilis* as compared with *E. coli*. Thus, we can conclude that antibacterial efficiency was found to be AgNPs concentration-dependent in the order S3 > S2 > S1 > S0 and maximum ZOI was noted in the case of sample S3 that had a higher density of AgNPs as confirmed by SEM, XRD and UV–vis results.

### 3.7.2. Density-dependent bactericidal analysis

The actual mechanisms of antibacterial activity of NPs against different bacterial strains are not fully understood, but three different routes of bactericidal action are well known, such as the release of metal ion, non-oxidative process and oxidative stress induction. Owing to these simultaneously occurring

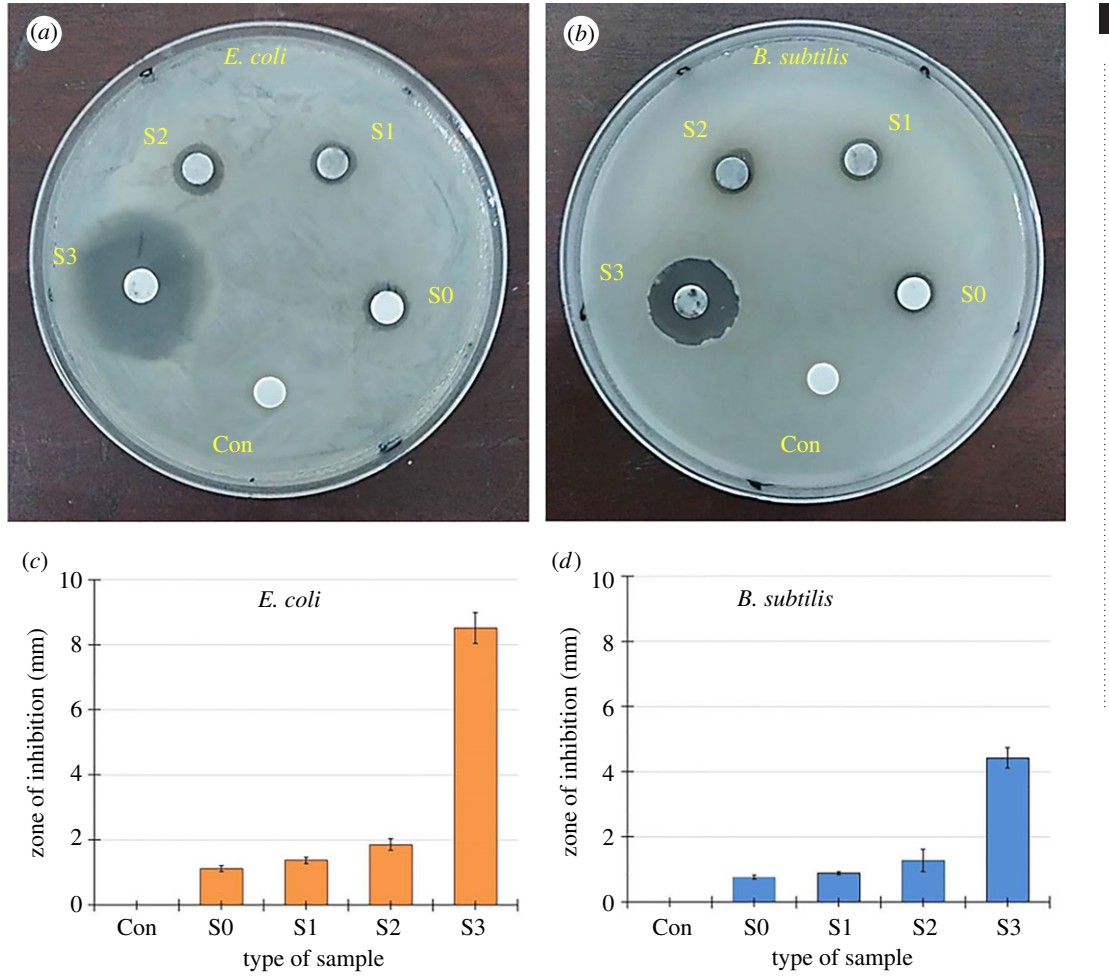

**Figure 8.** Representative images of (*a*) *E. coli* and (*b*) *B. subtilis* agar plates showing ZOI around discs impregnated in samples S0, S1, S2, S3 and pure water (Con). The graphical representation of all measured ZOI against (*c*) *E. coli* and (*d*) *B. subtilis*.

**Table 4.** Average zone of inhibition (ZOI, mm) of prepared samples against *E. coli* and *B. subtilis*.

| sample | E. coli | B. subtilis |
|---|---|---|
| Con | 0 | 0 |
| S0 | 1.12 $\pm$ 0.09 | 0.76 $\pm$ 0.06 |
| S1 | 1.37 $\pm$ 0.10 | 0.89 $\pm$ 0.47 |
| S2 | 1.86 $\pm$ 0.17 | 1.28 $\pm$ 0.34 |
| S3 | 8.51 $\pm$ 0.48 | 4.42 $\pm$ 0.32 |

biocidal activities of NPs, microorganisms fail to develop required multiple gene mutations for their survival and thus NPs-based actions are found effective against antibacterial-resistant microbes [60].

The potential antibacterial mechanism of nanocomposites (Co@AgNPs) against bacteria in the present study can be attributed to the interplay between AgNPs and that of the bacterial cell wall. This may work under the effect of electrostatic interactions, hydrophobic effects and van der Waals forces. This may lead to the adsorption of AgNPs on the cell membrane and production of silver ions (Ag$^{+}$). The released Ag$^{+}$ might have damaged the cell membrane by protein coagulation, cell wall pits, inactivation of the respiratory chain, membrane permeability induction and biosorption process. Furthermore, adhesion of AgNPs with cell membrane probably produces the reactive oxygen species (ROS) that damage the bacterial cell membrane. The leakage of cellular contents can lead to cell lysis [60,61]. Moreover, due to the induced permeability, small-sized AgNPs could have penetrated into the

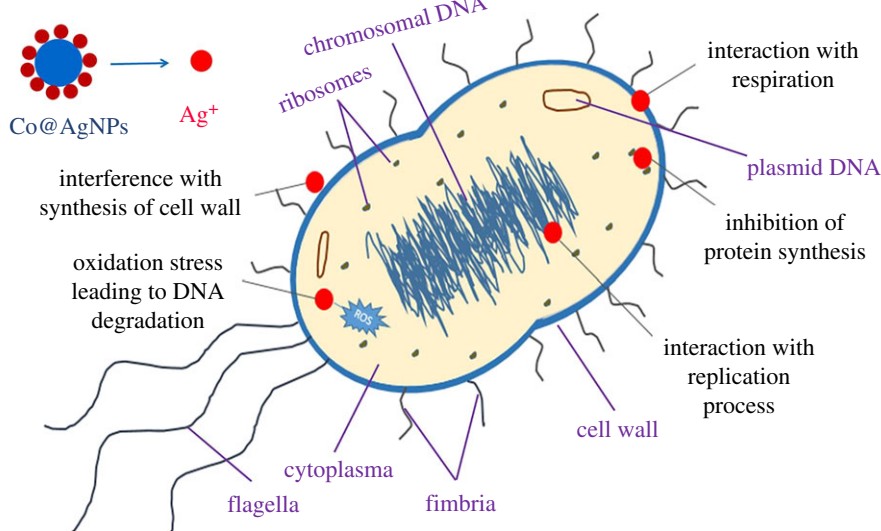

**Figure 9.** Potential bactericidal mechanisms of Co@AgNPs against bacterial cells.

bacteria. AgNPs interact with vital components of bacteria and can result in various damage, such as abnormal proteins and nucleic acid synthesis; destruction of cytoplasm membranes; abnormal DNA replication; inhibition of lysosomes, enzymes and ribosomes; disorders in electrolyte balance; and changes in gene expression leading to oxidative stress [2,60,62]. The magnetic nature of nanocomposites could further enhance the effectiveness of AgNPs by accumulation and aggregation within the organisms. The possible interactions of Co@AgNPs with bacterial strains and their effects are schematically demonstrated in figure 9.

The increased antibacterial activity of nanocomposites with a higher density of AgNPs (sample S3) revealed that perhaps higher concentration of AgNPs released more silver ions (Ag$^+$) which subsequently enhanced bactericidal performance against *E. coli* and *B. subtilis*. However, the antibacterial activity of prepared AgNPs was found to be higher for *E. coli* than that of *B. subtilis*. This is interesting because *E. coli* are more resistant than *B. subtilis* due to the presence of thicker membranes. These results are consistent with already reported data that AgNPs are a very efficient antibacterial agent against *E. coli* [2,61,63].

The magnetic behaviour and AgNPs density-dependent bactericidal efficiency of our bimetallic composite particles make them talented nano-carriers for drug delivery systems. By changing the density of decorated AgNPs, the available surface area and thus the amount of the loaded drug can be controlled. The magnetic nature can ensure the effective recovery of nanocomposite containers from the medium after successful drug transport to the target sites. We hope that the findings of this study could facilitate in designing a new and more effective paradigm of drug containers.

## 4. Conclusion

Bimetallic nanocomposites (Co@AgNPs) having magnetic cobalt cores decorated with a varied density of AgNPs were successfully synthesized. The enhanced density of AgNPs on the Co-cores was achieved by increasing the amount of silver precursors as 0.01, 0.1 and 0.2 g in reaction solution. The morphology, crystalline nature, magnetic and optical responses of manufactured particles were determined by SEM, XRD, VSM and UV–vis spectroscopy, respectively. The morphological analysis revealed that nanocomposite particles comprised magnetic sphere-shaped cores covered by spherical AgNPs of varied diameters (30–80 nm). The cores and loaded particles were of pure metallic (cobalt and silver, respectively) with crystalline (FCC) nature as confirmed by XRD. The optical characteristic indicated the increasing amount of silver content on cobalt cores and VSM spectra showed that cobalt cores retained their magnetic properties even after AgNP deposition. MTT assay showed that the prepared nanocomposite particles were not cytotoxic at low concentrations. The antibacterial ability of the prepared samples against *E. coli* and *B. subtilis* was found to be AgNP density-dependent. The nanocomposites with a maximum concentration of AgNPs showed the highest antibacterial efficiency against both bacteria. The manageable surface to volume ratio of these nanocomposites due to the

presence of AgNPs makes them more beneficial than core–shell structures. The synthesized hybrid nanocomposites with magnetic ability and tenable antibacterial performance can be considered as effective drug carriers.

Data accessibility. Data available from the Dryad Digital Repository at: https://doi.org/10.5061/dryad.0bk10qf [64].

Authors' Contributions. Z.K. and M.A.R. designed the experiments, analysed the data and wrote the manuscript. S.R. and S.N. conducted the characterization measurements and helped in the interpretation of data. S.M. and M.A.R. performed the synthesis experiments. Z.K., S.M. and I.S. performed the antibacterial activity experiments. A.T. conducted the cell cytotoxicity experiments and helped in data analysis.

Competing interests. We have no competing interests.

Funding. This work was financially supported by Higher Education Commission (HEC) of Pakistan under the Project of 'National Research Program for Universities' (project no. HEC-NRPU-4176).

Acknowledgements. The authors acknowledge Dr Muhammad Javaid Iqbal, Dr Shahid Atiq and Dr Syed Sajjad Hussain from Centre of Excellence in Solid State Physics, University of the Punjab, Lahore, Pakistan for their cooperation in experimental work and useful discussions in manuscript writing.

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
