## [Reviewer comments · Royal Society Open Science]

Review History

RSOS-182135.R0 (Original submission)

Review form: Reviewer 1

Is the manuscript scientifically sound in its present form?

Yes

Are the interpretations and conclusions justified by the results?

Yes

Is the language acceptable?

Yes

Is it clear how to access all supporting data?

Yes

Do you have any ethical concerns with this paper?

No

Have you any concerns about statistical analyses in this paper?

No

Recommendation?

Major revision is needed (please make suggestions in comments)

Comments to the Author(s)

Major: The advantage of the composite materials by using Co core has not been demonstrated.

Minor: Some of the data can be combined or put part of them into supporting information (e.g., Fig. 1-3; Fig. 6, etc).

Review form: Reviewer 2

Is the manuscript scientifically sound in its present form?

Yes

Are the interpretations and conclusions justified by the results?

Yes

Is the language acceptable?

No

Is it clear how to access all supporting data?

Yes

Do you have any ethical concerns with this paper?

No

Have you any concerns about statistical analyses in this paper?

No

Recommendation?

Major revision is needed (please make suggestions in comments)

Comments to the Author(s)

This manuscript described characterization, cell toxicity and anti-bacteria ability of cobalt cores functionalized with varied concentration of silver nanoparticles because magnetic cores loaded with metallic nanoparticles can be promising nano-carriers for successful drug delivery at infectious sites. Synthesized bimetallic nanocomposites (Co@AgNPs) showed expected features and abilities against bacteria. The work has been well done, and almost results are sound. However there are some points which the authors must address are shown as follows;

1. In Introduction, as the authors mentioned, did they believe that the Co@AgNPs administered can be collected really by magnetism?
2. Can we further compare antibiotics and some of nanoparticles in the same line or same level?
3. In Figs. 2 and 3, is there a Table or Figure on rice consumption?
4. Are there any data to prove the changes in color and the state that the authors explained?

5. In Fig. 3, the photographs presented are not in focus.
6. In Fig. 6, why is there no shoulder or peak showing silver nanoparticles at 400 to 500 nm?
7. In Fig.7, is the title on the horizontal axis correct?
8. As results of MTT assay, Co@AgNPs showed cytotoxicity from 5 μ g / ml (significant differences should be present between 0 of all and S0, S2 or S3). Why did the authors explain that Co@AgNPs can be safely used? And the authors must show significant differences among them in the figure.
9. There are some typographical and grammatical errors in the manuscript (For example, "respectively" must be ", respectively". Please check English throughout manuscript).

Decision letter (RSOS-182135.R0)

25-Feb-2019

Dear Dr Raza:

Title: Synthesis and characterization of silver nanoparticles decorated cobalt nanocomposites (Co@AgNPs) and their density-dependent antibacterial activity
Manuscript ID: RSOS-182135

The editor assigned to your manuscript has now received comments from reviewers. We would like you to revise your paper in accordance with the referee and Subject Editor suggestions which can be found below (not including confidential reports to the Editor). Please note this decision does not guarantee eventual acceptance.

Please submit your revised paper before 20-Mar-2019. Please note that the revision deadline will expire at 00.00am on this date. If we do not hear from you within this time then it will be assumed that the paper has been withdrawn. In exceptional circumstances, extensions may be possible if agreed with the Editorial Office in advance. We do not allow multiple rounds of revision so we urge you to make every effort to fully address all of the comments at this stage. If deemed necessary by the Editors, your manuscript will be sent back to one or more of the original reviewers for assessment. If the original reviewers are not available we may invite new reviewers.

Please also include the following statements alongside the other end statements. As we cannot publish your manuscript without these end statements included, if you feel that a given heading

is not relevant to your paper, please nevertheless include the heading and explicitly state that it is not relevant to your work.

• Ethics statement

Please clarify whether you received ethical approval from a local ethics committee to carry out your study. If so please include details of this, including the name of the committee that gave consent in a Research Ethics section after your main text. Please also clarify whether you received informed consent for the participants to participate in the study and state this in your Research Ethics section.

OR

Please clarify whether you obtained the necessary licences and approvals from your institutional animal ethics committee before conducting your research. Please provide details of these licences and approvals in an Animal Ethics section after your main text.

OR

Please clarify whether you obtained the appropriate permissions and licences to conduct the fieldwork detailed in your study. Please provide details of these in your methods section.

On behalf of the Subject Editor Professor Anthony Stace and the Associate Editor Professor Claire Carmalt.

RSC Associate Editor:
Comments to the Author:
(There are no comments.)

RSC Subject Editor:
Comments to the Author:
(There are no comments.)

Reviewers' Comments to Author:
Reviewer: 1

Comments to the Author(s)
Major: The advantage of the composite materials by using Co core has not been demonstrated.

Minor: Some of the data can be combined or put part of them into supporting information (e.g., Fig. 1-3; Fig. 6, etc).

Reviewer: 2

Comments to the Author(s)

This manuscript described characterization, cell toxicity and anti-bacteria ability of cobalt cores functionalized with varied concentration of silver nanoparticles because magnetic cores loaded with metallic nanoparticles can be promising nano-carriers for successful drug delivery at infectious sites. Synthesized bimetallic nanocomposites (Co@AgNPs) showed expected features and abilities against bacteria. The work has been well done, and almost results are sound. However there are some points which the authors must address are shown as follows;

1. In Introduction, as the authors mentioned, did they believe that the Co@AgNPs administered can be collected really by magnetism?
2. Can we further compare antibiotics and some of nanoparticles in the same line or same level?
3. In Figs. 2 and 3, is there a Table or Figure on rice consumption?
4. Are there any data to prove the changes in color and the state that the authors explained?
5. In Fig. 3, the photographs presented are not in focus.
6. In Fig. 6, why is there no shoulder or peak showing silver nanoparticles at 400 to 500 nm?
7. In Fig.7, is the title on the horizontal axis correct?
8. As results of MTT assay, Co@AgNPs showed cytotoxicity from 5 $\mu\text{g} / \text{ml}$ (significant differences should be present between 0 of all and S0, S2 or S3). Why did the authors explain that Co@AgNPs can be safely used? And the authors must show significant differences among them in the figure.
9. There are some typographical and grammatical errors in the manuscript (For example, "respectively" must be ", respectively". Please check English throughout manuscript.

Author's Response to Decision Letter for (RSOS-182135.R0)

See Appendix A.

Decision letter (RSOS-182135.R1)

26-Mar-2019

Dear Dr Raza:

Title: Synthesis and characterization of silver nanoparticles decorated cobalt nanocomposites (Co@AgNPs) and their density-dependent antibacterial activity
Manuscript ID: RSOS-182135.R1

It is a pleasure to accept your manuscript in its current form for publication in Royal Society Open Science. The chemistry content of Royal Society Open Science is published in collaboration with the Royal Society of Chemistry.

On behalf of the Subject Editor Professor Anthony Stace and the Associate Editor Professor Claire Carmalt.

RSC Associate Editor
Comments to the Author:
The authors have addressed the reviewers comments well and the manuscript can be accepted.

Reviewer(s)' Comments to Author:

Appendix A

Reply to Reviewer's Reports

(Manuscript ID RSOS-182135)

We are thankful to reviewers for a vigilant assessment of our manuscript. We are grateful for their thoughtful and valuable comments to improve our manuscript. All points and issues raised by the learnt reviewers have been considered and the manuscript has been modified accordingly. A detailed point-by-point response to all comments is provided below indicating the implemented changes in in the revised version of the manuscript. A highlighted version by 'Track Changes' is also included with the resubmission.

Response to 1st Reviewer's Comments

Comment 1. Comments 1. Major: The advantage of the composite materials by using Co core has not been demonstrated.

Response 1. We are grateful to the learnt reviewere for highlighting this important point. The following paragraph has been added in the introduction to address this matter.

Among the magentic materials, Iron (Fe), Cobalt (Co) and Nickel (Ni) are considered most prominent and attractive ferromagnetic elements which belong to the 3d-block. Owing to their excellent magnetic characteristics in the elemental forms, they are widely used as magnetic cores for the fabrication of nanocomposite structures. For Fe, Co and Ni, the saturation magnetization (M_s) values are 220 emu/g, 170 emu/g and 55 emu/g, respectively at room temperature while Curie point (T_c) values are 770 °C, 1131°C and 358 °C, respectively. Although the higher M_s value of Fe makes it superior in magnetization in comparison to Co and Ni, the larger value of negative reduction potential (E^0) proves it unstable due to fast oxidation in air. This oxidation process at nanoscale further enhances due to larger surface to volume ratio making Fe-nanoparticles highly prone to oxidation as compared to other two elements. In the comparison of Co and Ni, pretty higher Curie point, larger magnetization and greater magnetocrystalline makes Co an ideal core material for hybrid nanostructures. Various nanocomposite materials including core-shell structures have been reported using the Co as core material, for example, cobalt-gold (Co@Au), cobalt-copper (Co@Cu), cobalt-platinum (Co@Pt) and cobalt-silver (Co@Ag) nanoparticles [R1-R4]. The prominent advantages of using Co as core material with other non-ferromagnetic materials such as Ag, Au and Cu

include giant magnetoresistance (GMR) and high stability even at higher temperatures which are desirable properties in multifunctional sensing, media recording and wideband photovoltaic solar cell applications [R5-R8].

Comment 2. Minor: Some of the data can be combined or put part of them into supporting information (e.g., Fig. 1-3; Fig. 6, etc)

Response 2. We are thankful to the reviewer for kind suggestions. Fig. 2 and 3 are combined as recommended by the reviewer. Fig. 1 depicts the whole experimental procedure and Fig. 6 (now Fig. 5) represents the optical analysis of prepared colloidal samples so they are kept in the main text. However, data of Fig. 6 (now Fig. 5) has been uploaded to Dryad Digital Repository <https://datadryad.org/review?doi=doi:10.5061/dryad.0bk10qf>

Response to 2nd Reviewer's Comments

Comment 1. This manuscript described characterization, cell toxicity and anti-bacteria ability of cobalt cores functionalized with varied concentration of silver nanoparticles because magnetic cores loaded with metallic nanoparticles can be promising nano-carriers for successful drug delivery at infectious sites. Synthesized bimetallic nanocomposites (Co@AgNPs) showed expected features and abilities against bacteria. The work has been well done, and almost results are sound. However there are some points which the authors must address are shown as follows;

Response 1. We are obliged to the reviewer for mentioning our work “*well done*” and our results “*sound*”.

Comment 2. In Introduction, as the authors mentioned, did they believe that the Co@AgNPs administered can be collected really by magnetism?

Response 2. We are grateful to the reviewer for pointing out this point. The manipulation of administered magnetic nanoparticles (MNPs) or composite MNPs by magnetism (external magnetic field or magnetic implants) is well

documented for different purposes including magnetic drug delivery, magnetic separation, magnetic resonance imaging and magnetic hyperthermia [R9-R11]. The in vivo movement of MNPs strongly depends upon the factors such as external magnetic field gradient, morphology of the MNPs, the temperature and viscosity of the medium because these factors control the interaction of fluid components with MNPs. By optimizing these governing parameters, the MNPs can be accumulated in a specific area or transported to some target site [R12]. In the case of our Co@AgNPs, since composite particles retained their magnetic properties as revealed by VSM spectra, we can believe that our composite MNPs can be manipulated and collected by magnetism.

Comment 3. Can we further compare antibiotics and some of nanoparticles in the same line or same level?

Response 3. Yes, we have already shown in our previous study [R13] that AgNPs exhibited an effective antibacterial performance comparable to antibiotics (Ciprofloxacin).

Comment 4. In Figs. 2 and 3, is there a Table or Figure on rice consumption?

Response 4. According to the reviewer's recommendation, the following table describing the quantitative details of all chemicals used in preparation of our samples shown in Fig. 2 and 3 has been added in the main text.

Table 1 Details of chemicals and experimental conditions for the preparation of composite structures (Co@AgNPs).

Sample	Precursor	Reducing agent	stabilizing agent	Reaction conditions
S0	Cobalt acetate (48mM, 25 ml)	Hydrazine hydrate (0.4 ml), Ethylene glycol (25 ml), Added dropwise	PVP (0.25 mM, 50 ml)	Heated up to 195°C temp under refluxing conditions with constant stirring (100 rpm)
S1	Silver acetate (0.01 g) added to S0	–	–	Mixed by stirring (100 rpm) at room temp
S2	Silver acetate (0.1 g) added to S0	–	–	Mixed by stirring (100 rpm) at room temp

S3	Silver acetate (0.2 g) added to S0	–	–	Mixed by stirring (100 rpm) at room temp.
----	--	---	---	---

Comment 5. Are there any data to prove the changes in color and the state that the authors explained?

Response 5. As the chemical reduction reaction proceeds, different phases of the reaction occur and reaction solution takes different colors. We took pictures of different stages occurring during the synthesis of particles. We only obtained the UV-Vis spectra of final stage of each sample. Similar explanations of color changes during synthesis of NPs have also been reported in other studies [R14-R16]

Comment 6. In Fig. 3, the photographs presented are not in focus.

Response 6. Fig. 3 (now Fig. 2 lower panel) has been replaced with better resolution images.

Comment 7. In Fig. 6, why is there no shoulder or peak showing silver nanoparticles at 400 to 500 nm?

Response 7. We are grateful to the reviewer for pointing out this matter. The absence of any sharp peak at 400 to 500 nm could be due to various factors, for instance, in the case of composite structure the peak position, peak width and peak intensity may be affected by the nature and amount of the component materials. Furthermore, the type of composite structure, for example, core-shell or cores decorated with NPs may also affect the optical properties. Different studies has reported the change in position and shape of absorbance bands due to the core-shell structures, for example, red-shift and band broadening in case of Co@Au [R17] and Co@Ag [R18] while blue-shift and increase in bandwidth for Ni@Ag [R19] and Au @ Ag [R20]. Also absence of any sharp absorbance peak was reported in case of Fe₃O₄ cores decorated with AgNPs [R21].

In our case, we noticed different absorbance behavior for S2 and S3 as compared to S0. The broad bands starting from 330 nm and centered at 430 nm can be attributed to the presence of AgNPs on the Co-cores. This band broadening may be ascribed to size distribution of both core and decorating NPs, synergetic phase retardation effects and electron scattering at the sharing interfaces [R22, R23]. Similar broad absorbance bands without any sharp shoulder were reported by Lee et al. for AgNPs decorated on Fe₃O₄ cores [R21]

while Fageria et al [R24] also observed the band widening and reduction in the sharpness of the absorbance peaks for ZnO/Ag and ZnO/Au nanoparticles. The lack of uniformity in NPs-content deposited on the cores in the case of NPs decorated core configurations unlike the core-shell structures where uniform and thick shells-content are present may also contribute to the band broadening [R25] and thus causing the reduction of peak sharpness.

A paragraph has also been added in section “optical characterization analysis” of the main text to address this point.

Comment 8. In Fig.7, is the title on the horizontal axis correct?

Response 8. The title on the horizontal axis of Fig. 7 (now Fig. 6) has been corrected.

Comment 9. As results of MTT assay, Co@AgNPs showed cytotoxicity from 5 $\mu\text{g} / \text{ml}$ (significant differences should be present between θ of all and S0, S2 or S3). Why did the authors explain that Co@AgNPs can be safely used? And the authors must show significant differences among them in the figure.

Response 9. The degree of biocompatibility and biosafety (%) would be relative and depends on kind of bioapplications. According to the ISO norm, a material is to be considered biocompatible (nontoxic) and safe for bioapplications if the cell viability is not below than 70% [R26]. In our nanocomposite (Co@AgNPs) particles, cell viability was found to be more than 70% up to the concentration of 10 $\mu\text{g}/\text{ml}$ ($74.45 \pm 2.3\%$, $90.59 \pm 1.0\%$, $81.83 \pm 2.5\%$, $75.06 \pm 1.4\%$ for S0, S1, S2 and S3, respectively) as shown in Fig. 8 (now Fig. 7). Thus we can say that the prepared Co@AgNPs can have potential for bioapplications

According to the reviewer`s recommendations the significant differences among all samples at different concentrations have been added in figure 8 (now Fig. 7).

Comment 10. There are some typographical and grammatical errors in the manuscript (For example, “respectively” must be “, respectively”. Please check English throughout manuscript.

Response 10. We thank the reviewer for advising to correct the typographical and grammatical errors which will enhance the beauty of our manuscript. All the required corrections were made accordingly.

References

- [R1]. Bao Y, Krishnan KM. 2005. Preparation of functionalized and gold coated cobalt nanocrystals for bio medical applications. *J. Mag. Mag.Mater.* **293**, 15-19. (doi: 10.1016/j.jmmm.2005.01.037)
- [R2]. S.S. Kalyan Kamal SSK, Sahoo PK, Sreedhar B, Raja MM, L. Durai L, Ram S. 2012. Chemical synthesis of Co/Cu core/shell nanocomposites and evaluation of their magnetic properties. *Mater. Sci. Eng. B.* **177**, 1206-1212 (doi: 10.1016/j.mseb.2012.06.001)
- [R3]. Garcia-Torres J, Valles E, Gomez E. 2010. Synthesis and characterization of Co@Ag core-shell nanoparticles. *J. Nanopart. Res.* **12**, 2189–2199. (doi: 10.1007/s11051-009-9784-x)
- [R4]. Xiang T, Wan J, Liu L, Gao JJ, Xu HT, Zhang HJ, Gu X, Wang Y. 2018. Thickness-tunable core-shell Co@Pt nanoparticles encapsulated in sandwich-like carbon sheets as an enhanced electrocatalyst for the oxygen reduction reaction. *J. Mater. Chem. A.* **6**, 21396-21403. (doi: 10.1039/C8TA05114C)
- [R5]. Garcia-Torres J, Valles E, Gomez E. 2012. Measurement of the giant magnetoresistance effect in cobalt-silver magnetic nanostructures: nanoparticles. *Nanotechnology* **23**, 405701(8pp). (doi: doi:10.1088/0957-4484/23/40/405701)
- [R6]. Coey JMD, Fagan AJ, Skomski R, Gregg J, Ounadjela K. 1994. Magnetoresistance in nanostructured Co-Ag prepared by mechanical-alloying. *IEEE Trans. Magn.* **30**, 666–668. (doi: 10.1109/20.312370)
- [R7]. Yuping Bao Y, Calderon H, Krishnan KM. 2007. Synthesis and Characterization of Magnetic-Optical Co-Au Core-Shell Nanoparticles. *J. Phys. Chem. C.* **111**, 1941-1944. (doi: 10.1021/jp066871y)
- [R8]. Yujun S, Jie D, Yinghui W. 2012. Shell-Dependent Evolution of Optical and Magnetic Properties of Co@Au Core-Shell Nanoparticles. *J. Phys. Chem. C.* **116**, 11343-11350 (doi: 10.1021/jp300118z)
- [R9]. Kudr J, Haddad Y, Richtera L, Heger Z, Cernak M, Adam V, Ondrej Zitka O. 2018. Magnetic Nanoparticles: From Design and Synthesis to Real World Applications. *Nanomaterials*, **7**, 243. (doi:10.3390/nano7090243).
- [R10]. Price PM, Mahmoud WE, Al-Ghamdi AA, Bronstein LM. 2018. Magnetic Drug Delivery: Where the Field Is Going. *Front. Chem.* **6**, 619. (doi: 10.3389/fchem.2018.00619).
- [R11]. Kolosnjaj-Tabi J, Lartigue L, Javed Y, Luciani N, Pellegrino T, Wilhelm C, Alloyeau D, Gazeau F. 2016. Biotransformations of magnetic nanoparticles in the body. *Nanotoday* **11**, 280—284. (doi: 10.1016/j.nantod.2015.10.001).
- [R12]. Colombo M, Carregal-Romero S, Casula MF, Gutiérrez L, Morales MP, Bohm IB, Heverhagen JT, Prosperi D, Parak WJ. 2012. Biological applications of magnetic nanoparticles. *Chem. Soc. Rev.* **41**, 4306–4334. (doi: 10.1039/c2cs15337h).
- [R13]. Raza RA, Kanwal Z, Rauf A, Sabri AN, Riaz S, Naseem S. 2016. Size and shape-dependent antibacterial studies of silver nanoparticles synthesized by wet chemical routes. *Nanomaterials* **6**, 74.(doi:10.3390/nano6040074).
- [R14]. Kanwal Z, Raza RA, Rauf A, Manzoor F, Riaz S, Ghazala Jabeen G, Fatima S, Naseem S. 2019. A Comparative Assessment of Nanotoxicity Induced by Metal (Silver, Nickel) and Metal

Oxide (Cobalt, Chromium) Nanoparticles in *Labeo rohita*. *Nanomaterials* 2019, 9, 309. (doi:10.3390/nano9020309).

- [R15]. Venkatakrisnan S, Veerappan G, Elamparuthia E, Veerappan A. 2014. Aerobic synthesis of biocompatible copper nanoparticles: promising antibacterial agent and catalyst for nitroaromatic reduction and C–N cross coupling reaction. *RSC Adv.* 4, 15003–15006. (doi: 10.1039/c4ra01126k)
- [R16]. Kharat SN, Mendhulkar VD. 2016. Synthesis, characterization and studies on antioxidant activity of silver nanoparticles using *Elephantopus scaber* leaf extract. *Mater. Sci. Eng. C. Mater. Biol. Appl.* 62 (2016) 719–724. (doi: 10.1016/j.msec.2016.02.024).
- [R17]. Bao Y, Calderon H, Krishnan KM. 2007. Synthesis and Characterization of Magnetic-Optical Co-Au Core-Shell Nanoparticles. *J. Phys. Chem. C* 2007, 111, 1941-1944. (doi: 10.1021/jp066871y)]
- [R18]. Gracis-Torres J, Valles E, Gomez E. 2010. Synthesis and characterization of Co@Ag core-shell nanoparticles. *J. Nanopart. Res.* 12, 2189–2199. (doi:10.1007/s11051-009-9784-x)]
- [R19]. Jing JJ, Xie J, Chen GY, Li WH, Zhang MM. 2015. Preparation of nickelsilver coreshell nanoparticles by liquid-phase reduction for use in conductive paste. *J. Exp. Nanosci.* 10(17), 1347-1356. (doi: 10.1080/17458080.2015.1012751)
- [R20]. Pyne P, Sarkar P, Basu S, Sahoo GP, Bhui DK, Bar H, Misra A. 2011. Synthesis and photo physical properties of Au @ Ag (core @ shell) nanoparticles disperse in poly vinyl alcohol matrix. *J. Nanopart. Res.* 13, 1759–1767. (doi: 10.1007/s11051-010-9955-9).
- [R21]. Lee B, Koo S. 2012. Silver reduction on the surface of magnetite nanoparticles using a coupling agent. *J. Ind. Eng. Chem.* 18, 1191–1195. (doi:10.1016/j.jiec.2012.01.009).
- [R22]. Song Y, Ding J, Wang Y. 2012. Shell Dependent Evolution of Optical and Magnetic Properties of Co@Au Core Shell Nanoparticles. *J. Phys. Chem. C* 2012, 116, 11343-11350. (doi:10.1021/jp300118z)
- [R23]. Westcott SL, Jackson JB, Radloff C, Halas NJ. 2002. Relative contributions to the plasmon line shape of metal nanoshells. *Phys. Rev. B.* 66, 155431. (doi: 10.1103/PhysRevB.66.155431)
- [R24]. Fageria P, Gangopadhyay S, Pande S. 2014. Synthesis of ZnO/Au and ZnO/Ag nanoparticles and their photocatalytic application using UV and visible light. *RSC Adv.* 4, 24962-24972. (doi: 10.1039/C4RA03158J)
- [R25]. Lyon JL, Fleming DA, Stone MB, Schiffer P, Williams ME. 2004. Synthesis of Fe oxide core/Au shell nanoparticles by iterative hydroxylamine seeding. *Nano Lett.* 4, 719–723. (doi: 10.1021/nl035253f)
- [R26]. International Organization for Standardization, ISO 10993-5:2009 Biological Evaluation of Medical Devices Part 5: Tests for in Vitro Cytotoxicity, 2009. ([nhiso.com/wp-content/uploads/2018/05/ISO-10993-5-2009.pdf](https://www.iso.com/wp-content/uploads/2018/05/ISO-10993-5-2009.pdf))